# Lignin Nanoparticles and Their Nanocomposites

**DOI:** 10.3390/nano11051336

**Published:** 2021-05-19

**Authors:** Zhao Zhang, Vincent Terrasson, Erwann Guénin

**Affiliations:** Université de Technologie de Compiègne, ESCOM, TIMR (Integrated Transformations of Renewable Matter), Centre de Recherche Royallieu, CS 60 319, 60 203 Compiègne CEDEX, France; zhao.zhang@utc.fr (Z.Z.); v.terrasson@escom.fr (V.T.)

**Keywords:** lignin nanoparticles, nanocomposites, preparation methods, biodegradability, non-toxicity, added-value applications

## Abstract

Lignin nanomaterials have emerged as a promising alternative to fossil-based chemicals and products for some potential added-value applications, which benefits from their structural diversity and biodegradability. This review elucidates a perspective in recent research on nanolignins and their nanocomposites. It summarizes the different nanolignin preparation methods, emphasizing anti-solvent precipitation, self-assembly and interfacial crosslinking. Also described are the preparation of various nanocomposites by the chemical modification of nanolignin and compounds with inorganic materials or polymers. Additionally, advances in numerous potential high-value applications, such as use in food packaging, biomedical, chemical engineering and biorefineries, are described.

## 1. Introduction

With our continued in-depth understanding of the environmental pollution and resource crisis, the renewable and degradable properties of biomass materials are being increasingly valued [1,2]. As the second most abundant natural polymer material after cellulose, lignin has received extensive attention in recent years [3,4]. The development of bio-based products from lignin is an important part of any comprehensive biorefinery concept because of their biocompatibility and biodegradability [5]. They not only diversify the combination of products and markets, but also benefit waste recycling and economic sustainability [6,7]. Nevertheless, worldwide only 5% of the lignin is explored for high value development, therefore, there are still numerous challenges and opportunities for the in depth research and development of lignin applications [8].

The three-dimensional network structure of lignin is formed by three phenylpropane monomers (*para*-coumaryl alcohol, sinapyl alcohol and coniferyl alcohol, Figure 1) which connect to each other through ether bonds and carbon-carbon bonds [9,10]. It contains multiple active functional groups, such as aliphatic, aromatic, hydroxyl groups, etc. The isolation technology of cellulose, hemicellulose and lignin in biomass is directly related to the effective utilization of biomass. The complexity and diversity of lignin structures mainly depend on their different sources, types, extraction and purification methods [11]. Different extraction methods and pulping procedures will produce lignin with different structures and properties, which determine its subsequent development and applications [12].

The conventional separation and extraction methods for lignin mainly include grinding, acid/alkaline/thioacid hydrolysis, cellulose enzymolysis, organic solvent extraction and ionic liquid extraction [13,14]. Organic solvent pulping reduces the dependence on alkali or inorganic acids in the traditional pulping process [15]. Organic solvents such as alcohols, esters and amines can be used to dissolve the lignin in the raw materials to achieve the purpose of separation from cellulose [16].

In addition, biological enzymes are used to selectively degrade cellulose to achieve lignin separation. This biological treatment separates the lignin with less impact on its structure and chemical properties [17]. Therefore, it is particularly important to select suitable and efficient extraction methods for different raw materials without destroying the lignin structure. More importantly, the presence of different types of ions and the ionic strength of solutions is the foundation for the association and isolation of lignin [18]. The macromolecular and colloidal properties of lignin can be investigated in depth by studying the self-aggregation kinetics at some specific solution conditions [19]. The study of lignin solutions/colloidal behavior and the associative/dissociative processes also facilitates the understanding of its physicochemical properties [20].

Wide applications of lignin as an additive in composite materials and bio-based products from lignin deconstruction are being intensively developed [21,22]. However, it is not possible to completely achieve the high-value applications of lignin due to some unfavorable byproducts. The diversity of lignin sources and the complexity of the structures make the research on the potential applications of lignin have certain opportunities and challenges [23,24]. Therefore, the formation of lignin-based nanomaterials will open up a different perspective for expanding the high-value applications of lignin.

As for other materials and biopolymers, such as cellulose and chitosan, the rationale for the preparation of lignin nanomaterials is to gain new attractive properties occurring only when matter is organized on a nanoscale [25]. This can be due to the so-called “quantum effect” bringing new tunable properties at the nanoscale or simply by the expanded surface to volume ratio of nanomaterials [26]. Compared with traditional composite materials, nanocomposites have obvious advantages, especially new materials from biorenewable and sustainable sources [27]. Due to the compatibility, degradability and environmental friendliness of bionanocomposites, their potential applications in the food packaging industry and pharmaceuticals are being exploited [28]. Polymer nanocomposites are being further used in additive manufacturing technology to produce more complex and diverse parts and components [29]. In addition, the application of multifunctional nanocomposite materials in the field of optics is also becoming increasingly prominent [30]. In the last few years, the research on multifunctional nanofibers is helpful for the development of potential applications in the field of medicine, biological tissue engineering, etc., especially chitosan electrospun nanofibers [31]. Electrospun nanofibers can be used to reinforce composite materials due to their specific molecular orientation and excellent mechanical properties [32].

As for lignin, with its aromatic and highly cross-linked network structure and chemical complexity, lignin contains plentiful of functional groups that are accessible for further surface modification [33]. This laat asset increases the potential activity of nanolignin to achieve high value-added applications of lignin [4]. In addition, the utilization of economical and environmentally friendly nanolignin as feedstock for the evolution of chemical industry conforms to green chemistry principles and sustainable development concepts [13].

Currently, the exploitation of nanolignin is the subject of a tremendous amount of research [34]. Lignin nanoparticles with different morphologies (smooth colloidal, hollow, spherical and quasi-spherical, Figure 2) have been successfully synthesized by controlling the reaction conditions of solvent/anti-solvent, the lignin concentration, the temperature and pH of solution, etc. [35,36]. Lignin nanoparticles have potential applications in antioxidants, thermal/light stabilizers, reinforced materials and nanomicrocarriers owing to their advantages of non-toxicity, environmental resistance, excellent thermal stability and biocompatibility [37].

In recent years, the valorization of lignin, lignin nanoparticles and their nanocomposites have been extensively reviewed. Figure 3 displays the number and subject areas of published literature on lignin nanoparticles and nanocomposites over the last five years. Tetyana et al. [38] focused on lignin-inorganic composite materials and applications in energy storage. Mishra et al. [36] focused on non-covalent interactions and self-assembly of lignin, briefly describing synthetic methods without mentioning specific applications. Österberg et al. [39] provided only an overview of spherical lignin nanoparticles and their applications in dispersants, coatings, adhesives and composites. Duval et al. [40] described the structure and extraction process of lignin, as well as the applications of lignin-based polymeric and micro/nano-structured materials.

Therefore, in this review we would like to provide an outlook on this promising research on nanolignin and its nanocomposites. We will emphasize lignin nanoparticle synthesis methods and the characterizations of their structure and performance, along with the effects of different reaction conditions on the morphology and dispersion of lignin nanoparticles. Furthermore, the modification of lignin nanoparticles and multifunctional lignin-based nanocomposites are described. The difference in the properties of the modified lignin nanoparticles is emphasized. Finally, we will provide a novel perspective to the value-added applications of lignin nanomaterials and many promising possibilities to improve biotechnological developments.

## 2. Synthesis Methods of Lignin Nanoparticles

The physical and chemical properties such as non-toxicity, corrosion and UV-resistance, antibacterial and anti-oxidation activity of lignin are attracting more and more attention [3,33]. Therefore, the use of low-cost and abundant lignin raw materials to prepare nanoparticles is an important aspect of expanding their high value-added utilization [41]. At the same time, finding simple, scientific and safe methods for preparing lignin nanoparticles is of great significance. The current preparation methods of nano-lignin mainly include anti-solvent precipitation, self-assembly, gradual addition and mechanical methods, etc [1,7,10]. Some emerging methods, such as ice segregation-induced self-assembly, aerosol-flow synthesis and electrospinning, are also being developed [35,42,43,44,45]. Figure 4 gives an overview of the synthesis and modification methods of nano-lignin. Lignin nanoparticles with different morphologies and sizes, which can be prepared through these different methods and conditions, will be of great value for their subsequent applications in different fields [46].

### 2.1. Anti-Solvent Precipitation

Whatever its provenance, lignin is generally insoluble in water or acidic solutions, but has good solubility in common organic solvents such as THF and DMF [8,10]. In the preparation of lignin nanoparticles, THF and DMF are commonly used as organic solvents to dissolve lignin due to their excellent performance and their lack of effect on the structure of lignin [36]. In addition, the method for preparing nano-lignin by dissolving and re-precipitating lignin has the advantages of being a relatively simple operation and requiring low level of equipment.

#### 2.1.1. Water as Anti-Solvent

According to the difference in the solubility of lignin in organic solvents and water, the nanoparticles are precipitated from the solution due to the decreased solubility after the introduction of water. Table 1 shows the preparation details and properties of lignin nanoparticles by using water as anti-solvent.

Lievonen et al. [47] prepared spherical lignin nanoparticles of about 200 nm in size by dialyzing a solution of softwood kraft lignin and THF in deionized water. The minimum average diameter appeared at 1 mg/mL lignin concentration and the particle size enlarged as the concentration increased until 10 mg/mL, which could be explained by nucleation-growth mechanism. The stability of the dispersion was excellent in pure water and NaCl solution in a wide pH range. This could be interpreted by the high negative zeta potential (−60 mV) allowing for the electrical double layer repulsion mechanism. The nanoparticles were more stable and it was easier to control their spherical shape, compared with others reports [48,49] using the same preparation method. However, this methodology presents some disadvantages: the difficulty to accurately control the size of nanoparticles and the utilization of THF organic solvent, this is a straightforward preparation.

Lintinen et al. [50] utilized a mixed solvent of tetrahydrofuran (THF), ethanol (EtOH) and water to dissolve soft wood kraft lignin. Colloidal spherical lignin particles around 200 nm with a zeta potential of −40 mV were generated after concentration and drying. The proof-of-concept process was designed to prepare colloidal lignin particles on an industrial scale, which included five steps: lignin dissolution, CLP generation, solvent evaporation, ultrafiltration and spray drying. The large-scale and closed cycle production of nano-lignin will benefit large potential applications of lignin. However, it is difficult to control energy consumption and nanoparticles yield due to the complexity and separability of the 5-step approach.

Chen et al. [51] obtained quasi-spherical lignin nanoparticles around 100 nm by introducing deionized water into lignin dispersed aqueous sodium *p*-toluenesulfonate (pTsONa) solution. Various types of lignin (kraft lignin, sulfonate lignin and alkaline lignin) could be completely dissolved in the pTsONa solution at room temperature. The nanoparticle diameter could be controlled by varying the pH of the solution. The size of nanoparticle decreased as the pH value increased, which could be explained by the synergistic dissociation of pTsONa and the phenolic OH and COOH functional groups of lignin nanoparticles. This method avoids some limitations of the solubility of lignin species and the use of organic solvents. Nevertheless, the revealed irregularity of nanoparticle morphology and the instability in different pH solutions are drawbacks.

Li et al. [52] produced spherical hollow nanocapsules of around 63 nm size via self-assembly by adding water to a simple mix of kraft lignin/ethanol solution. The diameter of the nanocapsules increased as the concentration of lignin increased and the speed of water addition decreased. The mechanism of π-π interaction among the aromatic groups was suggested during the nanocapsules formation, which was confirmed by ultraviolet and infrared spectroscopy. Despite the lack of studies on nanocapsules stability and the pH effects on solution, the use of green solvents and the simple operation are obvious advantages.

In addition, some other organic solvents were employed to dissolve various lignin for preparation of lignin nanoparticles. Camargos et al. [53] and Yearla et al. [54] utilized a solution of acetone/water to dissolve lignin extracted from corn biomass, hardwood lignin and softwood alkali lignin, respectively. The spherical lignin nanoparticles around 100 nm were obtained by controlling the solution pH and dropping double-distilled water rapidly. High purity lignin was dissolved in acetone by Richter et al. [55] to achieve flash precipitation of lignin nanoparticles. Li et al. [56] published a method to prepare uniform nanospherical particles about 300 nm with hollow cavity space by dissolving kraft lignin in dioxane.

#### 2.1.2. Acid Solution as Anti-Solvent

The principle of acid precipitation for the preparation of lignin nanoparticles is similar to that of water as an anti-solvent, which is based on the difference in the solubility of lignin in acid solutions and organic solvents. Besides, according to the electrical double layer theory, lignin nanoparticles are easier to precipitate due to the large amount of H^+^ in the acid solution [57,58]. The preparation details and properties of lignin nanoparticles by using acid solution as anti-solvent are as shown in Table 2.

Richter et al. [34] synthesized the low-sulfonated lignin nanoparticles about 84 nm with a zeta potential of −33 mV by introducing HCl into a solution of lignin in ethylene glycol. The lignin nanoparticles were pH-stable and biodegradable. The nanocomposites of lignin nanoparticles and Ag^+^ were successfully prepared, which displayed more excellent antimicrobial activity than silver nanoparticles. The lignin nanoparticles (50–250 nm) were prepared by Gupta et al. [59] following the same method, as shown in Figure 5. Compared with original lignin, improvements in crystallinity and thermal stability were revealed by X-ray diffraction analysis and DSC analysis. A similar approach was employed by Yang et al. [57] to prepare lignin nanoparticles around 50 nm with different HCl concentration. The nanoparticles were more uniformly distributed and stable than original lignin in a wide pH range. Furthermore, another approach of HNO_3_ precipitation from NaOH aqueous solution was used by Frangville et al. [60] which prepared nanoparticles stable only at pH below 5. The excellent degradability and environmental compatibility were confirmed through dispersing lignin nanoparticles with microalgae and yeast.

Richter et al. [55] obtained kraft lignin nanoparticles (45–250 nm) through HNO_3_ flash-precipitation from ethylene glycol solution. The surface of the nanoparticles was coated with a cationic polyelectrolyte, which made their surface properties adjustable and increased their stability in high pH system.

Beisl et al. [61] designed three different precipitation setups (batch, T-fitting, static mixer) with different mixing speeds to generate lignin nanoparticles, introducing H_2_SO_4_/H_2_O solution into ethanol aqueous mixture. The smallest nanoparticles (almost 100 nm) could be produced by static mixer setup with the highest mixing speed. The molecular weight and chemical structure of lignin nanoparticles did not change during the precipitation process.

#### 2.1.3. Supercritical CO_2_ as Anti-Solvent

Supercritical flow technology has been widely used in the field of preparing nanoparticle materials on account of the many resulting unique physical and chemical properties [62,63]. The principle of supercritical antisolvent precipitation technology is that the solubility of lignin in supercritical fluid is less than the solubility in a solvent [64,65]. The solubility of lignin in the original solvent is reduced when the supercritical fluid is dissolved into the solution. A large degree of saturation is formed in a short time to precipitate high-purity lignin nanoparticles. The advantages of supercritical method are that the prepared nano-lignin has small particle size and narrow distribution due to the low viscosity and zero surface tension of supercritical fluid. Furthermore, the commonly used CO_2_ is non-toxic and inexpensive [66,67]. Table 3 indicates the preparation details and properties of lignin nanoparticles by using supercritical CO_2_ as anti-solvent.

Myint et al. [68] successfully prepared the quasi-spherical lignin nanoparticles (38 nm) through introducing compressed CO_2_ into kraft lignin/DMF solution, which exhibited high monodispersity and uniform size. The size of nanoparticles increased as the temperature increased and the pressure decreased. Furthermore, the influence of two different solution flow rates (0.03 and 0.06 kg/h) on the size was discussed: lower flow rates showing an increasing trend on nanoparticle diameter. These effects on the nanoparticles formation were attributed to the change in solubility between DMF and CO_2_. Moreover, the nanoparticles exhibited excellent properties, for instance, favorable thermal degradation, outstanding dispersion stability, excellent UV absorption and non-cytotoxicity.

A similar process with supercritical CO_2_ was used by Lu et al. [69] to prepare spherical nanoparticles around 144 nm in size. Lignin was dissolved in an acetone solution and CO_2_ was added at 35 °C and 30 MPa. No change in the amorphous chemical structure was confirmed by FTIR analysis and XRD analysis in the supercritical antisolvent process. The nanoparticles possessed significantly improved solubility in water and antioxidant activity as a result of the enhanced specific surface area, compared with original lignin.

The above three anti-solvent precipitation methods are often used to prepare lignin nanoparticles, but there are some disadvantages. For example, lignin cannot be completely uniformly dispersed in the solvent, and agglomeration may occur in some cases. In the subsequent process, the solvent has to be removed by rotary evaporation or freeze-drying, which is unfavorable for precise control of the size and morphology of the nanoparticles. In order to satisfy the diversified requirements in different application fields, other preparation methods of lignin nanoparticles with various morphology and sizes attract the attention of research.

### 2.2. Physicochemical Method

The physicochemical preparation of lignin nanoparticles can avoid the use of corrosive chemicals or organic solvents and reduce the harm to the environment [1,7]. The process is mainly to prepare nano-lignin through the action of mechanical force or ultrasound, including ultrasonication, high shear homogenization method and a combination of multiple mechanical means, which can effectively avoid complicated subsequent treatment [10,70]. There are no obvious changes in the structure and functional groups of nano-lignin prepared by mechanical method, compared with the original lignin. The basic physical and chemical properties of the original lignin are preserved and higher reactivity is obtained. In addition, the advantages of simple implementation and strong controllability of the physicochemical method are more prominent.

#### 2.2.1. Ultrasonication

Ultrasonication is already used for the synthesis of nanoparticles from biopolymers such as cellulose and chitosan [71,72]. Ultrasonic cavitation causes polymer decomposition, chemical bond breakage and free radical formation due to the high temperature and pressure generated during the cavitation process [73]. Ultrasonic method is a process of breaking lignin molecular bonds under ultrasonic energy [74]. The macromolecular lignin break down to form small molecular lignin, resulting in a reduction in the size of lignin particles [75,76]. Compared with the high shear homogenization method, the lignin nanoparticles prepared by the ultrasonic method have less sedimentation and better stability. The preparation and properties of lignin nanoparticles by ultrasonication and homogenization are as shown in Table 4.

Gilca et al. [77] obtained spherical lignin nanoparticles around 100 nm by ultrasonication of lignin/aqueous suspension at 20 kHz frequency for 60 min. The significant increase of the aromatic structure peak intensity and the aliphatic OH groups content was observed by FTIR and NMR spectroscopy, which could be explained by side chain cleavage after ultrasound treatment. In addition, the decrease in guaiacyl and p-hydroxy phenol groups as well as the average relative molecular weight was attributed to depolymerization during ultrasonic irradiation.

Solid and hollow spherical lignin colloids were synthesized through ultrasonic spray-freezing (1.7 MHz) of an alkali lignin/DMSO solution on liquid nitrogen cooled copper plate by Mishra et al. [78]. This process consisted of droplet and particle formations, which were determined by the frequency of ultrasonic nebulizer and the diffusional nature of solution. The hollow particles and solid ones were formed through peripheral or volumetric precipitation, respectively. The enhanced UV absorption ability because of layer by layer deposition was obviously observed.

Tortora et al. [79] developed ultrasound irradiation driven assembly of kraft lignin into spherical microcapsules (300–1100 nm) at the water/oil interface. The mechanism of microcapsules formation was explained by emulsification and cross-linking during the ultrasound treatment. Specifically, this was because ultrasound irradiation induced the formation of -OH radicals and the combination of -OH radicals and the phenolic groups on lignin. The excellent biocompatibility and non-cytotoxicity of lignin microcapsules were exhibited by encapsulation and further delivery of the hydrophobic molecule Coumarin-6 in vitro.

The alkali lignin nanoparticle dispersion (200 nm) and nanocomposite hydrogels were generated by Chen et al. [80] via ultrasonication using an ultrasonic Homogenizor at 25 kHz frequency for 1 h. Although there are no separated nanoparticles and no research on stability in different medium, the simple operation and environmental protection are obvious advantages.

#### 2.2.2. Homogenization

The high shear homogenization method is a process of dispersing and homogenizing the lignin particles in a suspension or emulsion system. Different from the ultrasonic method, the chemical bonds of lignin are broken under the action of high shear force. Nair et al. [81] presented a mechanical treatment to prepare the kraft lignin nanoparticles in deionized water by employing a high shear homogenization at 15,000 rpm for 4 h. No changes in chemical structure composition and molecular weight distribution, determined by NMR spectroscopy as well as GPC analysis, were observed after mechanical shearing.

The colloidal spheres lignin nanoparticles were generated by Rao et al. [82] via treatment of the organosolv lignin residues/ethanol/water mixtures with homogenization for 45 min. The more ethanol content in the solution, the more uniform the colloidal spheres size, which could be explained by the combination between the hydrophilic moieties of organosolv lignin residues and ethanol. Furthermore, the spheres diameters decreased from 300 nm to 170 nm as a result of the increase in homogenization time. Spherical cores and shells from the hydrophobic structures and hydrophilic moieties of lignin were formed through the Van der Waals and the π-π interactions during the homogenization.

### 2.3. Other Preparation

There are some disadvantages in the preparation of lignin nanoparticles by organic solvent-based self-assembly. For example, the additional separation steps and the treatment of residual organic solvents are required. The morphology and size of lignin nanoparticles are affected by the amount of organic solvents. In addition to the above-mentioned relatively mature methods for preparing nano-lignin, some new methods have been explored based on the characteristics of the raw materials selected and the specific applications of nano-lignin [7]. Table 5 displays the properties of lignin nanoparticles as well as the advantages and disadvantages of preparation methods.

#### 2.3.1. Ice Segregation-Induced Self-Assembly

Spender et al. [43] presented an ice-segregation-induced self-assembly process to produce kraft lignin nanofibers (diameter < 100 nm) in aqueous solution. Specifically, the lignin solution was dispersed on a steel surface tempered with liquid nitrogen to 77 K through syringe. The nanofibrous morphology and the phenomenon of priority alignment on the frozen front were obviously observed by SEM. The high and uniform freezing rates and low solution concentrations played an extremely important role in forming much smaller and more uniform nanofibers. Furthermore, the carbon nanofiber network structure was generated after carbonization of lignin fibers at 1000 °C. This reproducible methodology could be employed for preparation of carbon nanofibers from other water-soluble polymers.

#### 2.3.2. Aerosol-Flow Synthesis

Ago et al. [42] developed an aerosol-flow method to synthesize the spherical particles (30 nm–2 μm) with different lignins from the respective H_2_O or DMF solution via in situ size fractionation. The lignin droplets were transported to the laminar flow reactor by a nitrogen stream, and then dried to form solid particles. Although the fractional yields were affected by the solution concentrations and types of lignin, the total yield could reach more than 60%. The increase in size and narrower distribution of the lignin particle could be observed as the concentration of lignin solution increased. In addition, the lignin particles could be uniformly and stably re-dispersed in different oil/water media and polar/non-polar solutions under shear or heating, even in Pickering emulsions. This novel and high-yield method through aerosol flow reactor will benefit from large-scale production of lignin nanoparticles with controllable size and morphology.

#### 2.3.3. Electrospinning of Solutions

Electrospinning is a process of forming a jetstream of lignin solution under high-voltage electrostatic stretching and spraying from the spinneret hole to form polymer nanofibers [83,84,85]. During electrospinning, the positive electrode is placed on the syringe containing the lignin solution and the negative electrode is connected to a collection plate. When there is no external voltage applied, the lignin solution fluid forms droplets suspended at the needle mouth under the action of gravity and surface tension. When turning on high voltage, the droplets spray a fine stream from the needle under the action of electric field force. As the solvent evaporates, the fine stream solidifies on the collection device to form lignin fibers.

Ruiz-Rosas et al. [44] employed the electrospinning of Alcell lignin/ethanol solutions to generate lignin submicron fibers (400 nm–1 μm). A 76% yield of lignin fibers after the stabilization treatment and the 40% yield of carbon fiber after the carbonization treatment could be obtained, respectively. Weight-loss owing to the release of H_2_O and CO_2_ could be illustrated by the cleavage of hydroxyl and carboxyl groups in the lignin degradation. The smooth surface without any defects in lignin submicron fibers was characterized by SEM and TEM images. The microporous structure of the carbon fibers was confirmed by N_2_ adsorption-desorption isotherms at −196 °C. Furthermore, the carbon fibers possessed the favorable stability in air at low temperature and excellent oxidation resistance.

The similar electrospinning method was utilized by Ago et al. [45] to prepare lignin-based nanofibers reinforced with cellulose nanocrystals. The effect of electrical conductivity and surface tension of solutions on the electrospinnability and morphology of lignin-based nanofibers was investigated by SEM images. Additionally, the nanofiber composites possessed the enhanced thermal stability due to hydrogen bonding interaction between lignin and cellulose nanocrystals.

### 2.4. Synthesis of Modified Lignin Nanoparticles

The strong hydrogen bonding between the molecules is the main reason for the difficulty of dispersion and size reduction of lignin nanoparticles. The smaller the nanoparticles, the greater the surface energy. Lignin nanoparticles are easier to agglomerate due to the stronger hydrogen bonding effect. The reactivity and surface properties of lignin nanoparticles can be effectively improved through chemical modification [41,86]. The level of uniform dispersion can be increased by using an effective chemical modification to shield the hydrogen bonds effect so that the force between the lignin molecules is completely or basically just Van der Waals force [1,13]. Table 6 shows the synthesis of modified lignin nanoparticles by using different modifiers and methods.

Qian et al. [87] prepared acetylated modified lignin colloidal spheres (110 nm) using acetyl bromide as modifier and water precipitation method in THF. The acetylated lignin was more hydrophobic than the original alkali lignin as a result of the conversion of hydroxyl groups to ester groups. This would be more conducive to the formation of lignin colloidal spheres composed of hydrophobic cores and hydrophilic shells. The improved dispersibility of acetylated lignin colloids in organic solvents and water was exhibited due to amphiphilic functional groups in the colloidal spheres.

The CO_2_/N_2_ responsive dispersion/precipitation technology was developed to form alkali lignin nanoparticles modified by grafting of 2-(diethylamino)ethyl methacrylate [3]. Specifically, CO_2_ bubbling promoted the dissolution of modified lignin, while N_2_ bubbling facilitated the precipitation of modified lignin nanoparticles. Moreover, the CO_2_/N_2_ triggered dispersion/flocculation process was reversible. The faster dissolution and precipitation occurred as the graft density and chain length decreased. The lignin nanoparticles (200–500 nm) modified with cholesteryl chloroformate were prepared by Pourmoazzen et al. [88]. The high hydrophobicity of lignin nanoparticles was observed, which could be used to deliver folic acid.

The oil/water interface cross-linking polymerization method was employed to generate lignin-based spherical nanocapsules modified by grafting of allyl bromide and subsequent thiol-ene radical reaction [89]. The controlled release of hydrophobic coumarin-6 from the nanocapsules was investigated at different pH value. Nypelö et al. [90] utilized a similar method to fabricate submicron colloidals (90 nm–1 μm) modified through the crosslinking of epichlorohydrin and alkali lignin. The types and concentrations of surfactants and crosslinkers had a significant effect on the size and morphology of lignin colloids. The potential applications of lignin-based nanocolloid in drug delivery and organic carriers will be inspired by the miniemulsion polymerization approach.

Yiamsawas et al. [91] synthesized kraft lignin nanocarriers of various morphologies (solid, core-shell, porous) modified by esterification with methacrylic anhydride via free radical polymerization. A higher release of porous nanoparticles to hydrophobic cargo was observed because of their higher surface area. In addition, the addition of enzyme laccase could also promote the release of hydrophobic drugs. Hydroxymethylated modified lignin nanoparticles were fabricated through a HCl-precipitation method [92]. The optimized conditions for synthesis of nanoparticles were determined by the linear fitting regression equation.

**Table 6 nanomaterials-11-01336-t006:** Synthesis of Modified Lignin Nanoparticles.

Modifier	Methods	Lignin Source	Solvent	Diameter (nm)	Morphology	Mechanism	Reference
acetyl bromide	H_2_O- precipitation	alkali lignin	THF	110	colloidal spheres	self- assembly	[87]
(diethylamino) ethyl methacrylate	CO_2_/N_2_ -dispersion/ precipitation		DMF/ H_2_O	237–404	nanoparticle	dispersion flocculation	[3]
allyl bromide trimethylolpropane tris(3-mercaptopropionate)	mini emulsion polymerization	sodium lignosulfonate	butyl acetate/hexadecane/H_2_O	50–400	spherical nanocapsule	oil/water interface cross-linking	[89]
methacrylic anhydride		kraft lignin	DMF/ hexadecane	250–2000	solid core-shell porous	free radical polymerization	[91]
epichlorohydrin	microemulsification	alkali lignin	octane/H_2_O	90–1000	submicron colloidal	oil/water interface cross-linking	[90]
formaldehyde	HCl- precipitation	sarkand grass lignin	NaOH/H_2_O	200	nanoparticle	self-assembly	[92]

With the continuous in-depth research on the technology of preparing nano-lignin, it has been possible to obtain nanoparticles of various morphologies and sizes from different sources of lignin. In addition, these modified methods can realize the specific functionalization of lignin nanoparticles and give them some new chemical, optical and mechanical characteristics. The prepared lignin-based nanomaterials and products will also have higher value applications in chemical processing, biomedicine and food safety. Nevertheless, these preparation and modification methods still need to be optimized and improved to make the resulting nanoparticles more controllable and stable. Of course, searching for novel processes and scientific methods to prepare lignin nanoparticles has also become an urgent research focus.

## 3. The Value-Added Applications of Lignin Nanomaterials

In the process of exploring nanolignin materials, it was found that sometimes the ordinary nanolignin cannot satisfy the diverse requirements in different application fields. In addition to the basic thermal and mechanical properties of nanomaterials, some diverse nanostructures and specific performance such as metal adsorption are widely required [93,94]. Therefore, the ordinary nanolignin can be used to directly modify or compound with other nanomaterials, which can take advantage of nanolignin structure and further improve the performance of nanocomposite materials [95].

Lignin nanoparticles have attracted more and more attention because of their green, renewable and abundant source, antioxidant, antibacterial or ultraviolet absorption properties, biodegradability, biocompatibility, etc [96]. They are excellent substitutes for partially harmful nanomaterials, which are extensively used in the fields of drug release and control [97], food packaging, biomedicine, adsorbent materials, nanocarriers, environmental restoration and so on [98,99,100,101]. This not only solves the potential safety hazards of traditional nanomaterials from the source, but also broadens the value-added applications of lignin nanomaterials, which conforms to the principles of green chemistry development [102].

### 3.1. Antibacterial Effect

Lignin has antibacterial properties due to its benzene ring structure and phenol units. The inhibitory effect of phenolic unit on bacterial growth has been deeply studied by predecessors [103,104]. They found that the lignin extracted from different methods had different bactericidal properties, which was mainly due to the diversity of phenolic derivatives. Compared with the original lignin, nano-lignin has better antibacterial properties because of the increase in specific surface area and the number of phenolic side chains.

Silver nanoparticles have a wide range of antibacterial and antiviral activities [105]. They are widely used as bactericidal materials because they can effectively kill prokaryotic microorganisms, such as *Escherichia coli, Listeria monocytogenes* and *Pseudomonas aeruginosa* [106].

Richter et al. [34] developed lignin-based-Ag^+^ ions nanocomposites coated with a cationic polyelectrolyte in aqueous solution. The antibacterial experiments showed that the nanocomposites could kill common Gram-positive and Gram-negative pathogenic bacteria and quaternary amine-resistant bacteria. Furthermore, the amount of silver nanoparticles used in the nanocomposites was 10 times less than that used in traditional methods. This was attributed to the dual bactericidal properties of lignin nanoparticles and silver ions. However, the reduction of the silver ion content in the nanocomposite resulted in a decrease in the antibacterial effect. It was really necessary to solve the problem of silver ions release from lignin nanoparticles. The preparation of biodegradable nanolignin-silver ions nanocomposites will enlighten the emergence of more nanomaterials loaded with various metal ions.

The lignin nanoparticles are promising high-value biological additives for polymer nanocomposite films, which can be used in the field of advanced food packaging. The nanocomposite films of lignin nanoparticles (LNP), chitosan (CH) and polyvinyl alcohol (PVA) were successfully prepared by Yang et al. [107] through solvent casting. Two bacterial plant pathogens, *Xanthomonas arboricola* pv. pruni and *Pectobacterium carotovorum* subsp. odoriferum, were used to evaluate the antibacterial properties of the nanocomposite films. The results indicated that binary (PVA/LNP and CH/LNP) and ternary (PVA/CH/LNP) nanocomposite films could significantly inhibit the growth of bacterial plant pathogens. More importantly, the ternary composite system exhibited higher antibacterial activity than others. Although the addition of lignin nanoparticles reduced the transparency of nanocomposite films, it obviously increased thermal stability. Furthermore, Yang et al. [108] dispersed lignin nanoparticles in the nanocomposite films of cellulose nanocrystals (CNC) and polylactic acid matrix (PLA). Compared with PLA pure film, the higher antimicrobial activity of ternary composite films against the pathogen P. *syringae* pv. tomato ((Pst) was observed. The improved antibacterial properties of ternary polymeric films were attributed to the synergy effect of cellulose and lignin nanostructures.

A composite of hydroxymethylated and epoxidized modified lignin nanoparticles with copper and zinc was produced as agent for wood bioprotection by Gîlcă and Popa [109,110,111]. The birch veneer samples were impregnated with lignin solution and nano-lignin-based composite solution, and then embedded in the soil for 6 months. The biological stability and hydrophobic properties of the composite were evaluated by measuring the weight loss rate and contact angle of the samples. The result demonstrated that the nano-lignin-based composite had a better bactericidal and protective effects on the samples.

Zimniewska et al. [112] prepared nanolignin by an ultrasonic method, and then treated linen fabric with nano-lignin solution to produce multifunctional products. They found that the modified linen fabric had obvious antibacterial properties against eight kinds of bacteria including *Escherichia coli, Bacillus licheniformis* and *Micrococcus flavus,* etc. The use of silicone emulsion could achieve better adhesion of nano-lignin on the linen fabric without damaging the original physical properties. This method allowed the linen fabric to obtain antibacterial properties and UV protection that are beneficial to the human body.

### 3.2. Reinforcing Materials

Lignin nanoparticles have received extensive attention as nanofillers because of their low cost, low density, renewability, degradability and surface active properties. The effects of residual lignin on composition, structure and properties of cellulose fibrils and films were reported by Jiang et al. [113]. It was found that lignin improved the thermal stability and the hydrophobicity of cellulose fibril-based film. Furthermore, the effect of lignin on the physical, mechanical and surface properties of nanocellulose films was completely investigated by Rojo et al. [114] The mechanical properties of lignocellulose nanofibrils were maintained due to the uniform distribution of lignin.

Compared with lignin, nano-scale lignin particles possess higher softening temperature, and better thermal stability, which can be effectively used as natural fillers for nanocomposites [115]. Table 7 displays these parameters for mechanical properties and degradation temperature of different polymers and their nanocomposites. The given data shows that the thermal stability and mechanical strength of high polymers can be improved by using nano-lignin as reinforcing material in polymer matrices.

Binary blend films of polyvinyl alcohol (PVA) and chitosan have been extensively studied [115,116,117]. However, there were some disadvantages, for example, the reduction of deformability and crystallinity as well as the increase in rigidity. Therefore, in order to overcome these shortcomings, the composite films were fabricated by Nair et al. [81] through blending kraft lignin nanoparticles and PVA matrix in aqueous solution. The lignin nanoparticles possessed more stable dispersion than non-nano particles in the PVA matrix. Compared with pure PVA films, the degradation temperature of composite films increased from 262 °C to 382 °C. The significantly improved thermal stability of nanolignin-PVA blends was ascribed to the better dispersity of nanolignin in the PVA matrix. Cellulose nanofibrils (CNF) and lignin nanoparticles (LNP) were used as additives to prepare the composite films of hexagonal boron nitride nanosheet (BNNS) and PVA [118]. The mechanical strength and thermal stability of the bilayer composite film were significantly improved because of the synergistic effect of CNF and LNP. The composite hydrogels of lignin nanoparticles, cellulose nanofibrils and PVA exhibited high dynamic rheology properties as reported by Bian et al. [119] The self-recovery property and superelasticity of the hydrogels were observed as a result of the three-dimensional network structure. The addition of lignin nanoparticles transformed the ordered arrangement of molecules into random distribution under shear force, which was responsible for the enhanced viscoelasticity of composite hydrogels.

The composites films of lignin nanoparticles and polylactide (PLA) were prepared by Lintinen et al. [50] The lignin nanoparticles could be uniformly and stably dispersed in PLA matrix and no changes in size and zeta potential were observed. Moreover, the tensile tests confirmed that the composite films possessed enhanced mechanical properties due to the loading of nano-lignin. Yang et al. [120] compared the tensile properties of nanolignin/PLA bionanocomposites synthesized by melt extrusion and solvent casting. The results indicated that the tensile strength and modulus of melt-extruded films were higher than that of solvent-cast films. When the addition of nano-lignin was increased from 1wt.% to 3wt.%, the elongation at break of melt-extruded films enhanced while the tensile strength and modulus decreased.

Lignin nanoparticles and carbon fibers were used as additives to produce ternary hybrid nanocomposites in the poly(trimethylene terephthalate) (PTT) matrix by Gupta et al. [121]. Compared with pure PTT matrix, the tensile flexibility and impact performance of nanocomposites significantly improved due to the addition of 1.5wt.% nano-lignin. Additionally, the impact strength of the ternary hybrid nanocomposites was higher than that of nanolignin/PTT nanocomposites.

The extensive applications of phenolic foams, such as insulation materials and fireproof board, are limited by the low mechanical properties and obvious friability. Therefore, Orozco et al. [122] employed lignin nanoparticles as reinforcement materials to improve the mechanical properties of phenolic foams. The compressive modulus and strength of the composites reached 128% and 174%, respectively, compared with phenolic foams without lignin nanoparticles. The biodegradability and sustainability of lignin nanoparticles could expand the applications of phenolic foams comparing with other synthetic fibers.

In order to improve the thermal stability and mechanical properties of natural rubber and styrene butadiene rubber, Jiang et al. [123,124] introduced nano-lignin into the rubber matrix to prepare the nanocomposites, as shown in Figure 6. The nanocomposites of natural rubber and poly(diallyldimethylammonium chloride) modified colloidal lignin were successfully produced [123]. The significantly improved tensile strength and shear strength were observed as the content of nano-lignin in the rubber matrix increased. Compared with pure natural rubber, the increased decomposition temperature of the nanocomposites was attributed to the interaction of nano-lignin with the rubber matrix. After that, the lignin-based nanosheets were used to reinforce styrene-butadiene rubber [124]. The glass transition temperature and storage modulus of the nanocomposites obviously increased due to the excellent compatibility of nano-lignin and rubber matrix. Therefore, lignin nanoparticles are a renewable alternative to carbon black for preparing various rubber products.

**Table 7 nanomaterials-11-01336-t007:** Mechanical properties and degradation temperature of different polymers and their nanocomposites.

Reference	Polymer	Composition and Content	Tensile Strength (MPa)	Young’s Modulus (MPa)	Degradation Temperature TGA (T_max_ °C)
[81]	PVA	pure PVA 10 wt.% LNP	---	---	262 382
[118]	PVA	PVA + 4 wt.% CNF PVA + 4 wt.% CNF + LNP	30.62 34.98	---	250 400
[119]	PVA	PVA + CNF PVA + CNF + 4 wt.% LNP	---	---	329 459
[50]	PLA	PLA PLA + 1 wt.% LNP	50 30	1200 1300	---
[120]	PLA	PLA PLA + 1 wt.% LNP	44.3 48.7	1955.8 2153.2	351.1 346.5
[121]	bio-PTT	PTT PTT + 1.5 wt.% LNP 1.5 wt.% LNP + 7 wt.% carbon fibers	51.49 59.16 61.74	2058 2227 2309	427 447 542
[123]	Natural rubber	pure + 7 wt.% LNP	25.24 29.24	2.00 2.95	370.0 386.5
[124]	Styrene-butadiene rubber	pure + 10 wt.% LNP	2.73 14.14	0.80 1.47	365.3 375.7

Lignin can be used as a precursor to prepare carbon nanofiber materials because of its high carbon content [125,126]. Carbon nanoparticles obtained after the thermal stabilization and carbonization of lignin were used as reinforcing materials [127]. The influence of KOH content on the morphology and size of carbon nanoparticles during the dissolution and carbonization of lignin was investigated. When KOH concentration increased from 0 wt.% to 15 wt.%, the specific surface area of carbon nanoparticles first increased and then decreased.

Through high temperature carbonization process, the lignin-based nanobiochar was prepared as a renewable alternative to carbon black for reinforcing styrene-butadiene rubber [128]. The tensile behaviors showed that the strength and elongation at break of the composites were improved due to the loading of lignin nano-biochar, compared with the original rubber. The potential applications of carbon nanoparticles with specific structures in polymer nanocomposites could be exploited due to the renewability and compatibility of lignin.

### 3.3. Anti-Ultraviolet Effect

The functional groups such as phenol and ketone in lignin can effectively absorb a wide range of ultraviolet light and reduce ultraviolet radiation, which makes lignin have excellent antioxidant capacity and anti-ultraviolet properties [129]. Nano-lignin can replace some inorganic nanoparticles in some fields as a result of its renewability, environmental protection and degradability [130]. When mixed with different polymers, the lignin nanoparticles can uniformly be distributed in the polymer matrix [131]. Nanolignin-based composites can be used in the food packaging and cosmetics industry due to their excellent oxidation resistance and anti-ultraviolet properties.

#### 3.3.1. Food Packaging Films

The free radical polymerization and masterbatch processes were combined for the first time to produce the nanocomposites of lignin nanoparticles and poly(methyl methacrylate)(PMMA) [132]. The results revealed that the light transmittance of nanocomposites decreased significantly as the content of nanolignin increased from 0.5 wt.% to 4.5 wt.%. However, the transparency of the nanocomposite sheets also decreased due to the loading of nano-lignin.

Tian et al. [133] employed the solution-cast process to generate the lignin nanoparticles/PVA nanocomposite films, as shown in Figure 7. The nanocomposite films had excellent UV resistance because the phenolic hydroxyl and carboxyl groups of nanolignin could convert photon energy into heat and then release it. The improved antioxidant properties of nanocomposite films were attributed to the radical scavenging ability of lignin nanoparticles. In addition, the enhanced thermal stability of nanocomposite films was associated to the interfacial interaction and compatibility between nanolignin and PVA matrix.

The chemically modified polylactic acid films are becoming a promising alternative to petroleum-based plastics in the advanced food packaging field. Yang et al. [134] introduced lignin nanoparticles into the polylactic acid films grafted with glycidyl methacrylate by the masterbatch approach. The nanocomposite films had better UV resistance than pure films because of the ultraviolet absorption properties of nanosized lignin and the uniform dispersion in the polymer matrix. Although there was a significant deformation loss in the composite films due to the addition of nanolignin, this masterbatch approach had excellent processability for the formation of nanolignin-based composites.

#### 3.3.2. Sunscreens and Cosmetics

Long-term exposure of the skin to ultraviolet light will initiate oxidative stress reactions and produce a large amount of active substances, which will cause oxidative damage and cell metabolism disorder [135,136]. In order to prevent cell aging, chemical sunscreens are becoming more and more popular. Lignin can be used in the development of cosmetic sunscreens because of its excellent antioxidant and UV absorption capabilities. The natural polyphenol extract in lignin can not only effectively filter ultraviolet rays, but also repair damaged DNA [137].

The lignin-based broad-spectrum sunscreens were developed through introducing lignin colloidal spheres into pure skin cream as reported by Qian et al. [138]. Compared with original lignin [3], significantly increased UV absorption performance and antioxidant property of the sunscreens with lignin colloidal spheres were observed. The protective effect of phenolic hydroxyl groups in lignin on human skin was investigated by comparing the role of acetylated lignin and organosolv lignin in sunscreens. It was found that more phenolic hydroxyl groups possessed better sun protection effect.

The spherical lignin nanoparticles were isolated from elephant grass by antisolvent precipitation and then added to a neutral cream to produce lignin-based sunscreens [139]. Compared with commercial sunscreens, the creams with 10% lignin nanoparticles exhibited lower light transmittance. Lee et al. [140] mixed light-colored lignin nanoparticles from rice husks into a moisturizing cream to produce broad-spectrum sunscreens. The higher sun protection factor was demonstrated comparing with non-nanoparticles. In addition, the synergistic effect of lignin nanoparticles and organic UV-filter sunscreens enhanced the UV resistance performance. The applications of lignin nanoparticles in cosmetic sunscreens play an important role in the development of high-value lignin.

### 3.4. Nanocarriers

There are a large number of functional groups such as benzene ring, hydroxyl, carboxyl and methoxy group in the lignin structure [141]. The lignin is capable to be chemically modified by a variety of substances due to the presence of these functional groups, thereby greatly expanding the application fields of lignin [89]. The excellent biocompatibility and the great variety of nanostructures make lignin nanoparticles interesting carriers for drug transportation [4].

Chen et al. [51] investigated the loading and release capabilities of lignin nanoparticles (LNPs) for three different drugs, the slightly water-soluble gatifloxacin (GFLX), completely water-soluble doxorubicin hydrochloride (DOX.HCl) and water-insoluble doxorubicin (DOX). The encapsulation efficiency of the LNPs for three drugs could reach 37.4%, 42.6% and 90%, respectively. The excellent encapsulating capacity was ascribed to the intermolecular interactions between the LNPs and drugs. Furthermore, the DOX.HCl and GFLX-encapsulated LNPs exhibited better release ability than DOX-loading LNPs. In vivo antitumor assessment was performed through injecting the DOX-encapsulated LNPs into mice implanted with tumors. There were no obvious histological differences comparing with the experimental group injected with phosphate buffer saline (PBS). Furthermore, the significant inhibition effect on tumors growth was observed in the mice group injected with DOX-encapsulated LNPs.

The lignin-based nanocapsules possessed the exceptional loading capacity and pH-controllable release ability for hydrophobic coumarin-6 [89]. The encapsulation efficiency of the nanocapsules for coumarin-6 could reach 0.713 mmol/g in microemulsion medium. The cumulative coumarin-6 release rate was 40% and 60% under the conditions of pH 7.4 and pH 4.0, respectively. This was because of the cleavage of the acid-sensitive β-thiopropionate cross-linking bond and the destruction of nanocapsules shells under acidic condition. Similarly, Tortora et al. [79] investigated lignin microcapsules for storage and delivery capacity of coumarin-6. It was revealed that the addition of surfactant poly(ethylene glycol) could promote the encapsulation efficiency of the hydrophobic molecule. Furthermore, the sodium dodecyl sulfate (SDS) could also promote the release efficiency of coumarin-6 due to the increased solubility of lignin microcapsules in SDS solution.

Yiamsawas et al. [142] synthesized biodegradable nanocontainers of lignin nanoparticles and polyurea/polyurethane via interfacial polyaddition. The nanocontainers possessed stable dispersion in organic solvents and water for several months due to the high crosslinking effect of nanolignin and polyurethane/urea bonds. Furthermore, the fluorescent dye sulforhodamine was chosen as a hydrophilic molecule to load on the hollow lignin nanocontainers. The enzyme laccase was employed for the degradation of lignin-based nanocapsules shells to achieve the release of sulforhodamine.

The effect of three different morphologies lignin nanocarriers on the release capacity for UV-active cargo 2-propylpyridine was illustrated by Yiamsawas et al. [91]. The release efficiency of solid nanoparticles, porous structures and hollow nanocapsules reached 10%, 25% and 40%, respectively. In addition, a significantly increased release efficiency was observed due to the degradation of nanolignin in the presence of enzyme laccase. The drug delivery in biomedicine will benefit from the lignin-based nanocarriers with morphology-controlled release ability.

Three other drugs were used to determine the loading and release abilities of the lignin nanoparticles (LNPs) by Figueiredo et al. [143]. The poorly water-soluble sorafenib (SFN) and cytotoxic benzazulene (BZL) could be efficiently loaded on nanoparticles, whereas hydrophilic capecitabine (CAP) could not be loaded. The release efficiency of SFN-LNPs and BZL-LNPs both reached about 100% at pH of 7.4 through the enzymatic degradation of LNPs. It turned out that BZL-LNPs had an increased anti-proliferation effect on cancer cells comparing with the free BZL.

Sipponen et al. [144] fabricated renewable biocatalysts of hydrolases-cationic lignin nanosphere composites for butyl butyrate synthesis in aqueous medium. Firstly, the lignin nanoparticles were coated with hydrolytic enzymes (cutinase and lipase). Afterwards, the biocatalysts were synthesized through entrapment of hydrolases-coated nanospheres by calcium alginate hydrogel beads. It turned out that butyl butyrate in the water/hexane mixture under enzyme catalysis was effectively synthesized. The lignin-based biocatalysts without chemical modification will benefit aqueous ester synthesis.

The high specific surface area and porous structure of lignin nanoparticles make them excellent carriers for loading metal ions. The potential applications of lignin-based metal ion nanocomposites, such as biocatalysts and nano-adsorbents, will be further developed and utilized.

The structural diversity of metal-organic nanocomposites formed by lignin nanoparticles and iron isopropoxide in THF solution was investigated [145]. The different morphology, such as network structure, solid nanoparticles, uniform spheres and hollow nanospheres, was prepared under different hydrolysis reaction conditions. Moreover, the obviously higher magnetization of Fe(OiPr)_3_-lignin nanocomposites was exhibited, compared with Fe(OiPr)_2_-lignin nanoparticles. Metal alkoxides nanocomposites have been widely used in rubber composite materials, directional polymerization catalysts, ceramics and metal oxide films [146,147]. Ni nanoparticles were supported on lignin-based carbon nanofibers as catalysts to depolymerize lignin in supercritical ethanol/water [148]. The results showed that the catalysts accelerated the depolymerization of lignin and increased the yields of phenols and lignin fragments. The loading of Ni nanoparticles on nanofibers did not change significantly after three cycles of catalysis.

The nanocomposites of silica and lignin modified with 3-chloro-2-hydroxypropyl trimethyl ammonium chloride were synthesized through co-precipitation method [149]. The morphologies of core-shell structure and porous strawberry-like structure were presented according to the different degree of quaternization of lignin. Silica nanocomposites were widely used in electrode materials, photocatalysis, biomedicine and other fields [150,151]. Taking into account that lignin nanoparticles are inexpensive, degradable, renewable and non-toxic, it will open up more novel paths for the development of multifunctional nanostructured biomaterials.

### 3.5. Anti-Pollution Applications

Metal nanoparticles are widely used in catalyzing organic synthesis and antibacterial materials due to their catalytic activity and bactericidal effect [152,153]. However, they are considered potentially harmful to the environment because of long-term exposure and the difficulty of recycling [154,155]. Therefore, it is really important to choose the suitable adsorbents for loading heavy metal nanoparticles. Lignin nanoparticles are excellent candidates as nanocarriers because of their biodegradability and reproducibility [4,90,156]. Lignin-based metal nanocomposites are playing an increasingly important role in degrading dyes, treating industrial wastewater and other environmental protection.

In order to reduce the environmental impact of silver nanoparticles, Richter et al. [34] loaded silver ions on lignin nanoparticles coated with a cationic polyelectrolyte. The experimental results showed that the use of silver nanoparticles was reduced under the same sterilization effect. Meanwhile, these nanocomposites had little impact on the environment due to their biodegradable lignin-based core structure.

Nypelö et al. [90] employed lignin supracolloids as nanoreactors for silver nanoparticles in microemulsion via in situ reduction. When mixing NaBH_4_ and AgNO_3_ microemulsions, the silver particles would agglomerate and precipitate. It was difficult to control the size of silver particles and separate them. However, after introducing lignin supracolloids in the mixed microemulsions, the lignin-based silver nanocomposites formed.

The lignin nanoparticles were used as templates for the formation of hollow metal-polymeric capsules because of their cheapness and sustainability. The lignin nanoparticles were coated with Fe(III)-tannin composite films and then dissolved by organic solvents to form Fe(III)-phenolic nanocapsules [157]. The high performance of hollow capsules in degrading organic dyes through Fenton reaction was presented. Many organic compounds such as carboxylic acids, alcohols and esters could be oxidized by Fenton reaction, due to the strong oxidizing properties of the mixture of H_2_O_2_ and Fe(III) nanocapsules. The nanocapsules could be widely used in printing, oily and dyeing wastewater treatment because of the high ability to remove refractory organic pollutants.

Kraft lignin and phenolic resin (PR) nanosphere composites were synthesized in aqueous ethanol solution by hydrothermal process [158]. The specific surface area of the nanospheres has increased significantly after the introduction of 40% lignin in the phenolic resin. After that, palladium nanoparticles were supported on PR/lignin nanospheres, which exhibited excellent catalytic properties and recyclability. It was found that the catalyst composites could reduce the toxic Cr(VI) and degrade dyes methyl orange and Rhodamine B. Compared with the Pd@PR catalysts, the Pd@PR/lignin nanospheres showed better reducibility and degradation rate due to the higher loading of Pd nanoparticles. Furthermore, the nanocomposites of Pd nanoparticles and Fe_3_O_4_-lignin were used to catalyze the reduction of Cr(VI) and Suzuki-Miyaura reaction by Nasrollahzadeh et al. [159], as shown in Scheme 1 and Scheme 2. The catalytic efficiency of Pd@Fe_3_O_4_-lignin for the reduction of Cr(VI) was higher than other catalysts, and the reduction reaction time decreased as the amount of the nanocomposites catalysts increased.

Inspired by green mussel, a lignin-based nano-adsorbent was developed through the combination of lignin/dopamine and Fe_3_O_4_ nanoparticles via nano-precipitation methods by Dai et al. [160] The nanocomposites had more efficient adsorption capacity for Cr(III) because of the network structure of lignin and the presence of dopamine, compared with other biomass-based adsorbents. Furthermore, the magnetic responsiveness of Fe_3_O_4_ nanoparticles was responsible for the multiple reusability of the lignin-based nano-adsorbents. Considering the recyclability of lignin-based nano- adsorbents, they could be used to purify the wastewater from chemical industries and improve the environment.

### 3.6. Biocompatible Applications

Nanotechnology has been gaining more and more attention in the medical field and in vivo biology due to its numerous potential applications [161]. Nanolignin can be used to produce micelles or capsules attributed to the polar phenolic hydroxyl and aliphatic hydroxyl groups in its phenylpropane skeleton. Furthermore, the characteristics of no-cytotoxicity, biocompatibility and adsorption of biological macromolecules are outstanding, which have great potential in the pharmaceutical and biomedical fields [162].

The biocompatibility and biodegradability of lignin nanoparticles were evaluated via culturing Pseudomonas aeruginosa (PAO1) in vitro by Myint et al. [68]. The PAO1 exhibited a higher growth rate in the nanolignin solution than that in the phosphate buffer saline (PBS). The incubation time of PAO1 in the nanolignin solution was also significantly reduced. The PAO1 labeled with green fluorescent protein was evenly dispersed in the solution due to the excellent biocompatibility and non-cytotoxicity of lignin nanoparticles.

Tortora et al. [79] employed hamster ovary cells to evaluate the biocompatibility of lignin microcapsules in vitro. The homogeneous distribution of lignin microcapsules in the cytoplasm of ovary cells was observed. The results proved that lignin microcapsules were non-cytotoxic and biocompatible. Frangville et al. [60] cultivated chlamydomonas reinhardtii and saccharomyces cerevisiae in physiological solution and lignin nanoparticles dispersion to evaluate the cytotoxicity of nanolignin. Identically, there was no negative impact on the normal growth of cells, further indicating no cytotoxicity and excellent biocompatibility of lignin nanoparticles. Therefore, lignin-based nanocomposites could be an alternative to toxic nanomaterials in drug delivery and pharmaceutical preparations.

The nanocomposite hydrogels of lignin nanoparticles and polyacrylamide were prepared through in-situ free radical polymerization by Chen et al. [80]. The cell culture tests revealed that non-cytotoxic hydrogels could facilitate the growth and proliferation of human esophageal squamous carcinoma cell. The biorenewable hydrogels will have great potential in the fields of development of bioartificial tissue, clinical application, and so on.

Figueiredo et al. [143] prepared Fe(III)-nanolignin and Fe_3_O_4_-infused lignin nanocomposites. And then the interaction between plasma proteins and lignin-based nanocomposites was evaluated by culturing human plasma in vitro. The elevated zeta potential of the nanocomposites was observed because some proteins were adsorbed on the surface of the nanoparticles. The excellent cytocompatibility of lignin-based nanocomposites was revealed. However, there was slight toxicity to the cells due to the formation of hydrogen peroxide or superoxide radicals. Moreover, the superparamagnetic properties of the nanocomposites were presented after loading Fe_3_O_4_ on the lignin nanoparticles, which could be used in some biomedical fields such as magnetic resonance diagnosis and treatment.

The negative charges on the surface of nano-lignin not only prevent nanoparticles from agglomerating in aqueous solution, but also facilitate the adsorption of cationic polymers, which make the surface of lignin nanoparticles positively charged. Substances with negative charges such as amino acids or proteins can be adsorbed on the positively charged lignin nanoparticles, thereby improving the application value of lignin [97].

Nanocomposites of lignin nanoparticles coated with various proteins were prepared in aqueous media through adsorption-driven assembly method [163]. The morphology changes of nanocomposites and the interaction mechanism between nano-lignin and different proteins were investigated in detail. The random coiled proteins could be coated more uniformly and stably on the surface of the lignin nanoparticles due to their 3D fold structure, compared with globular proteins. Hydrogen bonding and hydrophobic interaction forces promoted the adsorption and dispersion of proteins on the surface of nano-lignin. The study of adsorption mechanism will expand the applications of advanced nanocomposites in artificial tissue repair, wound healing and other biomedical fields [164,165].

## 4. Conclusions and Future Perspectives

Overall, this review summarizes the current status of the preparation of nano-lignin by sedimentation, mechanical, self-assembly and stepwise addition polymerization methods. In order to provide a reference for the chemical deep processing of biomass resources and the development of nano-lignin, the application characteristics of nano-lignin in UV protection and anti-bacteria, nano-fillers and biomass-based carriers are also outlined. Through the specific analysis of the preparation methods and application status of nano-lignin, it can promote the further research of nano-lignin and the development of novel nano-lignin-based products. This is of great significance to the utilization and sustainable development of lignin.

The multiple structures and the diverse properties of lignin make the prepared nano-lignin more complicated, which brings challenges to the research of the preparation and performance of nano-lignin. Meanwhile, it provides broader prospects and opportunities for the multi-functional and multi-field applications of nano-lignin. Nanometersized lignin with high specific surface area and activity is a novel approach to achieve high value-added utilization of lignin. Compared with the research of inorganic nanoparticles and renewable nanocellulose, the preparation and application of nano-lignin are still in their infancy. The scale and industrialization of nano-lignin-based products will become an important aspect of future lignin research.

In view of the problems of complex process and toxic organic solvents in the preparation of lignin nanoparticles, methods such as electrostatic spinning, self-assembly, ultrasonication and homogenization can be utilized. In terms of multi-functional applications of lignin-based nanomaterials, the range of use can be expanded by improving their strength, electrical conductivity, thermal stability, crystallization performance, etc.

The following points need attention in the application of nanolignin materials. First of all, achieving uniform dispersions of nanolignin in composite materials is a difficult problem to solve. Lignin nanoparticles are extremely prone to agglomeration because of the high surface energy and the large number of hydrogen bonds and Van der Waals forces between the molecules. Therefore, completely solving the problem of particle agglomeration and achieving the monodispersion of nano-lignin are the key to fully exerting the nano-effect. Furthermore, the diversified morphology and size of lignin nanoparticles are prerequisites for high value-added and multi-field applications of lignin. To solve the interface and dispersion problems of nano-lignin materials, it is essential to find chemical methods to modify nano-lignin and also supplement with effective physical dispersion methods, such as mechanical stirring, ultrasonication and high shear homogenization. In addition, the amount of nano-lignin added, the type of treatment agent and dispersing equipment are all key factors that affect agglomeration, which need to be controlled and improved during the preparation of lignin-based nanomaterials.

The structural and functional properties of lignin determine its extremely promising applications in the field of biochemicals. Lignin and its derivatives have a wide range of functionalities and can be used as dispersants, adsorbents/desorbents, oil recovery aids, asphalt emulsifiers, etc. They can also be converted into aromatics, agrochemicals, polymers and high-performance materials. However, all these processes depend on improvements and innovations in the field of catalysis and product separation. Most importantly, to realize the full industrial potential of lignin, further refinement of biopulp technology is needed to achieve efficient separation of lignin and cellulose. It can be said that the contribution of lignin to sustainable human development lies in its ability to provide a stable and continuous source of organic matter. Thus it can truly guarantee sustainable green development and energy supply.

In addition, the antioxidant and anti-UV effects of lignin are its most prominent properties, so the application of lignin-based composites in food packaging and other fields has been increasingly developed. The lignin-based nanocomposites are also excellent carriers for a variety of metal ions and drug loading due to their unique nano-effects. They can be used as bio-nanocomposite catalysts and reducing agents for heavy metal ions, which are of great importance for environmental protection and wastewater treatment. Furthermore, the biocompatibility and non-toxicity of lignin are being intensively investigated.

Numerous scientific findings indicate that lignin-based nanocomposites have a very promising future in the biomedical field. Specific lignins have significant anti-lipid peroxidation and oxygen radical scavenging effects. Significant inhibitory effects of lignin on the central nervous system or on cancer cell proliferation can also be observed. Most importantly, lignin nanoparticles can be used as biological carriers for drug delivery and targeted drug release. The molecular expression, biocompatibility and cytotoxicity of lignin-based nanocomposites in cell lines have also been intensively studied. Conclusively, the development of novel biomass materials and products will be applied to the medical field and have a positive and effective impact on human life and health.

## Data Availability

The study did not report any data.

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
