# Peer review of "Lignin Nanoparticles and Their Nanocomposites"

_nanomaterials, 2021, doi:10.3390/nano11051336_

Round 1
Reviewer 1 Report
The review titled “Lignin Nanoparticles and Their Nanocomposites” reviews the different methodologies for the formation of nanoparticles from lignin and associated hybrid applied materials. This field of research has gained a rather large attention over the past few years and the amount of research outcomes is proportionally large. The valorization of lignin is a significant requirement to a functioning bioeconomy, which is of utmost importance to address current crisis. The authors comprehensively listed the mechanisms of formation of lignin nanoparticles, discusses valorization of lignins across a range of applications and closes the manuscript on a brief outlook. The manuscript is relatively well written and informative.
The most significant, and actually quite impactful, shortcoming of the manuscript is that lignin nanoparticles, the valorization of lignins, and their nanocomposites have been extensively reviewed recently. I include herein a list of such reviews since 2017: International Journal of Biological Macromolecules (2021); ChemSusChem 13.17 (2020): 4344-4355; Nanomaterials 9.2 (2019): 243; Green Chemistry 22.9 (2020): 2712-2733; Polymers 12.5 (2020): 1134; International Journal of Biological Macromolecules (2020); Green Chemistry 22.3 (2020): 612-636; The Canadian Journal of Chemical Engineering 97.11 (2019): 2827-2842; Biotechnology Advances (2020): 107685; Applied Sciences 10.13 (2020): 4626; Bioresource Technology Reports, 9, 100374; International journal of molecular sciences 18.11 (2017): 2367; Biotechnology for biofuels 10.1 (2017): 1-11; Materials Horizons 7.9 (2020): 2237-2257. The authors absolutely need to differentiate their review from the recent review literature, and more clearly and uniquely differentiate the value of their manuscript.
Beyond distinctiveness, the manuscript may be recommended for publication (at the discretion of the editor(s) in sight of the above information) for its qualities, such as its thoroughness, written quality, and organization.
The points below should also be considered:
- Please discuss the impact and value of the various streams to obtain nanoparticles detailed in section 2 on sustainability, feasibility, and valorization of lignins.
- Please critically rank by feasibility and impact, the applications introduced in section 3.
- The outlook would benefit from additional depth such as considerations of life-cycle assessments and other biorefinery-associated issues, which have been addressed in the associated literature (not necessarily for lignins specifically). Herein, a more quantitative/numerical outlook would highly increase the merits of this review.
Author Response
please find the reply in the attached file

Reviewer 2 Report
This manuscript is very well written and easy to comprehend and easy to read.
Kraft lignins are complex materials affected by their isolation and the pulping procedures and type of wood that have been pulped. I would like to see a brief introduction of the commercial isolation of kraft lignin. Reading the manuscript, you sense that the isolation is based on simple precipitation, which is part of the isolation, but lignin structures are important and have complex association behavior, see e.g., Dutta S. et al in ACS Symposium Series (1989), 397(Lignin), 155-76. Sarkanen, S. et al in Macrolecules (1987) 17, 2588, Lindstroem, T in Colloid & Polymer. Sci (1980) 258, 168, and Norgren et al in Langmuir (2002) 18, 2859 and from this list you understand that this has been investigated for a long time. The presence of type ions, ionic strength is fundamental for their association behavior and their isolation and should be considered. I would be happy to see such a part of that kind of background for this review.
I understand that lignins have become hot partly because of the nano hype but I would like to see why nanolignins are interesting in the various applications. Being nano-sized per se is not interesting but maybe that micro-lignin or so is much less interesting than nanolignins.
Example: Microfibrillated lignocellulose are nanocellulose with a lignin cover, conceived to improve the interaction between hydrophobic polymers and the less hydrophilic lignin, as compared to hydrophilic cellulose. This is something that I think should be part of the composite part, in the manuscript e.g., Jiang, Y et al in. Cellulose (2019) 26(3), 1577 and Rojo et al in Green Chemistry (2015) 17(3), 1853.
When it comes to the “Reinforcemnt part”, I don´t think lignins have a glass transition temperature (softening temperature is a better word) is the right word for non-structured polymers and I don´t think that lignins can be crystalline and from table 2 I can´t see that the mechanical properties of high polymer can “be significantly improved” by using nano-lignin.
Author Response

(The authors gave the same response as above.)

Reviewer 3 Report
This paper shows a good review of synthesis methods of lignin nanoparticles and the characterizations of structure and performance. There are some issues that need to address:
- I believe this review would benefit from a table that could compare and provide an overview of the discussed approaches. The table should include the advantages and limitations of each approach. Also, some indications about the lignin nanomaterials and the application methods of these approaches are an added value for readers.
- The language of the paper needs to be improved, as such, it is really difficult to read...
- The article needs more comprehensive comparison tables with high details.
- Introduction is written simply, most recent research and innovation in lignin nanoparticles, nanocomposites, and nanofibers performances should be reviewed to show the gap of knowledge. The introduction should be extended with recent research papers. The introduction should be rewritten to show the highlights and novelty of the work. This part is very poor.
- section of drawbacks and future could be increased quality of the manuscript.
- Similar reviews have been published recently (e.g., Beisl, S., Friedl, A., & Miltner, A. (2017). Lignin from micro-to nanosize: applications. International journal of molecular sciences, 18(11), 2367. and Duval, A., & Lawoko, M. (2014). A review on lignin-based polymeric, micro-and nano-structured materials. Reactive and Functional Polymers, 85, 78-96). It is recommended to add a statement to clearly separate the current work from these similar references and also define the review period (e.g. last five years). Also, prepare statistical data (such as the number of documents, document per country) about you used references by created databank such as Scopus, Google scholar, and web of science.
Author Response

(The authors gave the same response as above.)

Round 2
Reviewer 1 Report
The authors thoroughly addressed the concerns put forward and the manuscript should be considered for publication.
Reviewer 3 Report
Title: Lignin Nanoparticles and Their Nanocomposites
In this revised manuscript, the authors have made corrections according to referee comments. In my opinion, the manuscript in current form could be considered for acceptance.